# Empowering alcohols as carbonyl surrogates for Grignard-type reactions

Chen-Chen Li [1], Haining Wang[1], Malcolm M. Sim[1], Zihang Qiu [1], Zhang-Pei Chen[1], Rustam Z. Khaliullin[1] & Chao-Jun Li [1✉]

The Grignard reaction is a fundamental tool for constructing C-C bonds. Although it is widely used in synthetic chemistry, it is normally applied in early stage functionalizations owing to poor functional group tolerance and less availability of carbonyls at late stages of molecular modifications. Herein, we report a Grignard-type reaction with alcohols as carbonyl surrogates by using a ruthenium(II) PNP-pincer complex as catalyst. This transformation proceeds via a carbonyl intermediate generated in situ from the dehydrogenation of alcohols, which is followed by a Grignard-type reaction with a hydrazone carbanion to form a C-C bond. The reaction conditions are mild and can tolerate a broad range of substrates. Moreover, no oxidant is involved during the entire transformation, with only $H_2$ and $N_2$ being generated as byproducts. This reaction opens up a new avenue for Grignard-type reactions by enabling the use of naturally abundant alcohols as starting materials without the need for pre-synthesizing carbonyls.

[1] Department of Chemistry and FQRNT Centre for Green Chemistry and Catalysis, McGill University, 801 Sherbrooke Street West, Montreal, QC H3A 0B8, Canada. ✉email: cj.li@mcgill.ca

In tandem with the significant advancements of biological and pharmaceutical technologies, the role of organic chemists has evolved beyond the discovery of new chemical transformations. Developments such as rapid and direct late-stage functionalizations of large molecules have shown great potentials, with increased significance of organic reactions[1]. The Grignard reaction is a fundamental transformation in chemical synthesis and has been continuously developed over the past century. Its importance is attributed to the reaction's versatility and capacity to form C–C bonds, leading to the formation of secondary and tertiary alcohols[2–7]. A key limitation of this reaction, however, is its instability and broad reactivity. In addition, classical synthetic methods used to transform carbonyl compounds often requires the participation of oxidants, many of which are hazardous and have poor functional group tolerance[7]. Insofar, the Grignard reaction has typically been limited to early-stage construction instead of the direct late-stage modification of complex molecules or natural products.

In contrast, alcohols are among the most naturally abundant functional groups, which are commonly found in biomass and natural products. The direct transformation of alcohols into C–C bonds has been a long pursuit of synthetic chemists[8–11]. This type of transformation would be an especially vital tool for the late-stage functionalization of alcohol-containing natural products and pharmaceuticals. Furthermore, this type of transformation will contribute greatly to the future sustainability of chemical syntheses by minimizing the number of steps required (Fig. 1). Motivated by these potential benefits, we contemplated the possibility of using alcohols as surrogates of aldehydes and ketones for the Grignard-type reaction via the in situ formal "dehydrogenation" of alcohol catalyzed by transition metals[8]. Early extensive studies have shown that ruthenium(II) and other transition-metal complexes are efficient catalysts for the aerobic oxidation of alcohols to carbonyls[12–16], which indicates the potential for hydroxyl groups to act as carbonyl surrogates. This strategy, however, has been limited to the hydrogen-borrowing aldol reactions, Michael additions[17–22] and reductive aminations[23]. The use of alcohols as carbonyl surrogates for a Grignard-type reaction has never been successfully demonstrated. In order to successfully develop this reaction, two key challenges must be overcome: (1) the incompatibility of both the acidic alcohol proton and the oxidant with Grignard reagents[2], and (2) the possible transmetallation of organometallic reagents with ruthenium catalysts which in turn attenuates the activity of catalyst[24].

Hydrazones are known as halogen-free, easily accessible and traceless carbanion equivalents. Their use as "carbanions" in various reactivity has been developed by our group over the past several years[25–28]. Our early studies have shown that hydrazones react efficiently with carbonyls via a 1,2-addition catalyzed by ruthenium(II)-phosphine complexes, and such a reactivity can

even be successfully applied in synergistic relay reactions[29]. More importantly, unlike the classical Grignard reagents, hydrazones show unique tolerance towards acidic protons such as hydroxyl and amino groups. Herein, we report a unique Grignard-type reaction with alcohol as a carbonyl surrogate and hydrazones as carbanion equivalents using a ruthenium(II) catalyst (Fig. 2b).

## Results

**Exploration of the hydride acceptor-system.** A key step of this surrogate strategy was the in situ catalytic generation of carbonyls from alcohols through their "dehydrogenation". Based on previous studies from our group[30] and others[31], β-hydride elimination of alcohols could be efficiently catalyzed by Ru(II)-complexes with or without a stoichiometric amount of oxidants (hydride acceptors). The former has often been conducted under milder conditions, while the latter usually requires high temperatures with the use of special catalysts[32–35]. Thus, our investigation started by seeking a proper hydride acceptor. On the basis of our previous work[27,29], a ruthenium(II)-bidentate phosphine system was first explored by varying the type of oxidants (hydride acceptors) used (Table 1). We observed a 9% yield of the desired product in the absence of a hydride acceptor (Table 1, entry 1). Under such conditions, however, most of the alcohols remained unchanged while the hydrazone substrate had mostly undergone the competing Wolff–Kishner reduction (WK reduction), which led to an overall low efficiency for the Grignard-type C–C bond formation. Inspired by the aerobic oxidation of alcohols, we then tested the reaction under air and O₂ atmospheres, respectively; however, even lower yields were obtained in both cases (Table 1, entries 2 and 3). In the cases of NCS, isoprene and silver oxide as oxidants, no generation of the desired product was observed (Table 1, entries 4–6). These results indicated that either the ruthenium(II) catalyst or the hydrazone substrate was incompatible with strong oxidants. For this reason, weaker oxidants such as copper oxide and DCB (2,3-dichlorobutane) were then tested. As expected, the product yield showed a notable increase. The conversion efficiencies of the reactants, however, were still relatively low (Table 1, entries 7 and 8). Nevertheless, these results suggested that weak oxidants can be tolerated in this Ru(II)-bisphosphine system, albeit not significantly promoting the oxidation.

**Exploration of oxidant-free and acceptorless system.** The results above led us to consider an oxidant-free strategy in order to increase the efficiency of alcohol dehydrogenation, which will in turn allow the subsequent 1,2-addition of hydrazone by modifying the catalytic system. Upon carefully analyzing the results with the Ru(II)-dcypf system, we attributed the main reason for the low efficiency to the inefficient kinetics of the dehydrogenation process. The extensive studies on Noyori-type reactions have

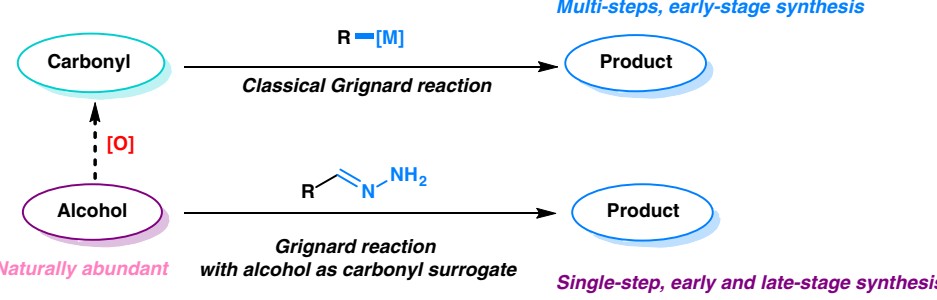

**Fig. 1 Classical Grignard reaction and Grignard reaction with alcohol as carbonyl surrogate.** This figure shows the comparison between classical Grignard reaction and the Grignard reaction with alcohol as carbonyl surrogate. It clearly shows that the latter is both step and atom economical. More importantly, using alcohol as carbonyl surrogate has a broad potential for late-stage functionalizations in synthetic chemistry.

**Fig. 2 Grignard-type reaction with alcohol as carbonyl surrogate. a** Classical Grignard-type reaction uses carbonyls as starting materials which come from the oxidation of alcohols. Organometallic reagents, in this case, serve as carbanion equivalent. **b** Alcohol-surrogated Grignard-type reaction, reported in this article, directly uses alcohol as starting material with hydrazones as carbanion equivalent, which skips the process of oxidation and generates hydrogen gas and nitrogen gas as only byproducts.

shown that a mixture of phosphine and amine ligands could accelerate the hydrogenation and dehydrogenation processes due to the favored six-membered pericyclic transition state[36]. Furthermore, most reported reactions concerning acceptorless dehydrogenation of alcohols require P-N type ligands[32–35]. These studies inspired us to investigate alternative catalytic systems other than Ru(II)-bisphosphine.

We started by using a well-defined Noyori-type ruthenium complex, Ru(dppf)(en)Cl₂ (dppf = diphenylphosphinoethane; en = ethylenediamine). In addition, we increased the hydrazone substrate equivalence to 3.5 in order to minimize the Wolff–Kishner reduction of the hydrazone. This initial attempt, however, only increased the yield slightly (Table 2, entry (1). We reasoned that since Ru(dppf)(en)Cl₂ is a stable complex with a limited number of empty coordination sites, it cannot drive the whole cascade process to proceed efficiently. To solve this problem, PNP-pincer type ligands[19,37–39] were then considered because: (1) with a tridentate structure, they can form more stable complexes with the metal center in order to stabilize the intermediates in the catalytic cycle, and (2) by occupying only three coordination sites, more space could be provided for the substrates and intermediates in order to facilitate the catalytic process. To our delight, the use of PNP-pincer type ligands indeed significantly increased the reactivity, among which bis[(2-diisopropylphosphino]ethyl)amine (**L3**) provided the best result (Table 2, entry 4).

Although the yield was increased, there were still some notable side products being generated when using **L3** as a ligand. One of the side products was the olefination product (**3aa-s1**) and the other was the hydrogen-borrowing hydrazination product (**3aa-s2**). Additionally, the Wolff–Kishner reduction also consumed all the remaining hydrazone before the complete conversion of the alcohol. We therefore carried out further optimizations in order to reduce the generation of these side products (Table 2, entries 5–9). The results showed that, by diluting the solution to 0.5 mL, the side reactions (i.e., the WK reduction and hydrogen-borrowing hydrazination) were significantly reduced without

impeding the reactivity of the desired reaction. Furthermore, the dilution also enabled the alcohol to be fully consumed in the presence of a smaller amount of hydrazone. Under the optimized conditions, different pincer ligands such as PNN-type and PN_pyridineP-type ligands were investigated; however, none of them gave better results compared to **L3** (Table 2, entries 10–12). Furthermore, pre-synthesizing the ruthenium complex with **L3** gave almost the same results as the one generated in situ (Table 1, entry 13). In addition, we also tested a less bulky PNP ligand, bis [(2-diethylphosphino]ethyl)amine (**L-Et**), and produced a well-defined complex (**Ru-PNP-2**) to run the reaction (Table 2, entry 14). The result with this complex was similar to the Ru-**L3** system. Later substrate scope studies also showed that both catalytic systems worked efficiently for primary alcohols. Thus, due to the fact that **L-Et** was less available, **L3** was used as the optimized ligand in most of our later studies. The use of a less bulky pincer ligand, however, did increase the yield for bulkier secondary alcohols, which will be discussed later.

**Investigation of the substrate scope.** With the optimized reaction conditions in hand, the substrate scope investigation (Fig. 3) was started for the alcohol partners in which simple alcohols were tested first. The results showed that the linear aliphatic alcohols examined all underwent the reaction smoothly. Notably, longer aliphatic chains led to lower yields (**3aa-3ac**); however, the overall yields were generally high. Aliphatic alcohols substituted with methylthio (**3ae**) and -NBoc (**3ah**) were also compatible with this process and provided moderate to high yields of the desired products. Similar results were obtained for heterocyclic substituted alcohols (**3ad-3ai**). In order to investigate the potential application of this reaction for pharmaceutical or agricultural industries, we demonstrated that the fluorine-containing alcohol also underwent the Grignard-type reaction and provided a moderate yield (**3hj**). A noteworthy finding was that small molecular alcohols such as ethanol could also participate in this C–C bond formation process at an elevated temperature (**3hk**).

**Table 1 Investigation of oxidants and hydride acceptors[a].**

| Entry | Oxidant | Result | Entry | Oxidant | Result |
|---|---|---|---|---|---|
| 1 | None | 9% | 5 | Isoprene | 0 |
| 2[b] | Air | Trace | 6 | Ag$_2$O | Trace |
| 3[c] | O$_2$ | 0 | 7 | CuO | 15% |
| 4 | NCS | 0 | 8 | DCB | 21% |

[a]Reaction conditions: **1a** (0.25 mmol), **2a** (0.2 mmol), Ru(PPh$_3$)$_3$Cl$_2$ (3 mol %), dcypf (3 mol %), K$_3$PO$_4$ (1.1 equiv), oxidant (2.0 equiv), solvent (0.2 mL) at 70 °C under N$_2$ atmosphere. See Supplementary Information (SI) for details. $^1$H NMR yield was determined using mesitylene as an internal standard. A 'trace' amount of product was noted when the desired product was not clearly detected.
[b]The reaction was performed under air.
[c]Oxygen balloon was used to provide O$_2$ gas during the reaction.

Benzyl alcohol and its derivatives were also effective substrates for this reaction; however, they generated more olefination products (**3am-3ao**). Increased steric effects suppressed this reaction as shown by the use of α-substituted alcohols (**3ap-3ar**) with the exception of α-cyclobutyl alcohol (**3ap**). A likely explanation was that the highly strained and small cyclobutyl group reduced the steric bulk around the metal center, generating the product in a relatively higher yield. In order to better illustrate the potential application of this reaction for total synthesis and late-stage functionalization, certain substrates containing sensitive functional groups were investigated. Substrates bearing amides and esters were well tolerated (**3ax**, **3az**), while the ones bearing more reactive functional group such as carbonate (**3ay**) demonstrated a lower yield. Nitriles and nitro-containing substrates were not competible, possibly due to their strong coordinating or oxidation ability.

Additionally, we noticed that the reaction with secondary alcohols both required harsher conditions and produced the corresponding products in relatively lower yields. The result further confirmed the significant steric effect of this reaction. To overcome this challenge, we switched the ligand from PNP **L3** to the less bulky **L-Et**. To our delight, when conducting the reaction under the catalysis of the ruthenium(II)-**L-Et** complex (**Ru-PNP-2**), the tested secondary alcohols (**3as-3hw**) reacted as efficiently as the primary ones, with the exception of **3au** due to its very high steric hindrance.

Subsequently, we decided to vary the hydrazones. Para-substituted benzaldehyde hydrazones were explored first, all of which produced the desired products in moderate to high yields. The CF$_3$ substituted hydrazone demonstrated the lowest yield due to the competing and rapid WK reduction in the presence of the strong electron-withdrawing effects of CF$_3$ (**3ba-3ea**). Similarly, most ortho- and meta-substituted benzaldehyde hydrazones proceeded smoothly and produced the desired products in moderate yields. Notably, certain hydrazones with low solubility in the reaction solvent, such as naphthaldehyde hydrazone (**3ha**), p-phenylbenzaldehyde hydrazone (**3ba**) and p-benzyloxylbenzaldehyde hydrazone (**3da**), were still able to undergo this transformation smoothly. Aliphatic aldehyde hydrazones proved to be much less reactive (**3ks-3ls**).

To further evaluate the application potential of this transformation, some naturally occurring complex alcohols, such as β-Citronellol and (−)-Nopol (**5aa-5ab**), were examined (Fig. 4). Both of them provided the desired Grignard-type reaction products in good yields. More importantly, the π-bonds in these natural products were unaffected during the reaction process. The olefin isomerization product as reported in our earlier studies[28] was not observed. A possible reason for the complimentary reactivity could be that in the PNP-Ru(II) system, the H$_2$ gas release proceeded much faster than the hydride insertion process. These results demonstrated a great synthetic value for C–C bond construction using olefinic natural alcohols, in which the chemoselective Grignard-type reaction of alcohols over olefin transformations could be realized.

**Mechanistic studies**. A tentative mechanism for this alcohol-surrogated Grignard-type reaction is proposed in Fig. 5 based on previous literature[19,27,29,36–41] as well as experimental results. The ruthenium(II) catalyst first coordinates with the PNP-pincer ligand **L3** to form complex **a** with the assistance of a base in order to form a highly reactive square planar complex[40]. The alcohol then interacts with complex **a** to undergo a β-hydride elimination via a Noyori-type six-membered-ring transition state **b** and produces the intermediate **c**[36]. This process is also supported by the Density Functional Theory (DFT) calculations (see SI for details). Next, the hydrazone substrate coordinates with the

**Table 2 Investigation of reaction conditions under oxidant-free system[a].**

| Entry | Catalyst | Ligand | K₃PO₄ (equiv) | Solvent (mL) | Yield |
|---|---|---|---|---|---|
| 1 | Ru(dppf)(en)Cl₂ | — | 2 | 0.2 | 22 |
| 2 | Ru(PPh₃)₃Cl₂ | L1 | 2 | 0.2 | 40 |
| 3 | Ru(PPh₃)₃Cl₂ | L2 | 2 | 0.2 | 66 |
| 4 | Ru(PPh₃)₃Cl₂ | L3 | 2 | 0.2 | 73 |
| 5 | Ru(PPh₃)₃Cl₂ | L3 | 1.5 | 0.2 | 66 |
| 6 | Ru(PPh₃)₃Cl₂ | L3 | 2 | 0.3 | 71 |
| 7 | Ru(PPh₃)₃Cl₂ | L3 | 2.5 | 0.3 | 64 |
| 8 | Ru(PPh₃)₃Cl₂ | L3 | 2 | 0.5 | 70 |
| 9[b] | Ru(PPh₃)₃Cl₂ | L3 | 2 | 0.5 | 74 |
| 10[b] | Ru(**PNN-1**)(PPh₃)Cl₂ | — | 2 | 0.5 | 36 |
| 11[b] | Ru(PPh₃)₃Cl₂ | L4 | 2 | 0.5 | 32 |
| 12[b] | Ru(**PNN-2**)H(CO)Cl₂ | — | 2 | 0.5 | 17 |
| 13[b] | **Ru-PNP-1** | — | 2 | 0.5 | 73 |
| 14[b] | **Ru-PNP-2** | — | 2 | 0.5 | 74 |

[a]Reaction conditions: **1a** (0.7 mmol), **2a** (0.2 mmol), Ru(PPh₃)₃Cl₂ (5 mol %), ligand (5 mol %), K₃PO₄, 2-Me-THF at 70 °C under N₂ atmosphere. See Supplementary Information (SI) for details. ¹H NMR yield was determined using 1,3,5-trimethoxylbenzene as an internal standard. A 'trace' amount of product was noted when the desired product was not clearly detected.
[b]**1a** (0.6 mmol) was used.

**Fig. 3 Substrate scope of the alcohol-surrogated Grignard-type reaction.** Reaction conditions: **1a** (0.6 mmol), **2a** (0.2 mmol), Ru(PPh₃)₃Cl₂ (5 mol %), ligand (5 mol %) and K₃PO₄ (2 equiv) in 2-MeTHF at 70 °C under N₂ atmosphere for 24 h. See Supplementary Information (SI) for details. Yield of isolated product was reported otherwise noted. **a** Ru-PNP-1 was used as catalyst. **b** ¹H NMR yield was determined using mesitylene as an internal standard. **c** The reaction was conducted at 100 °C, with **1a** (0.8 mmol) and Ru-PNP-2 being the catalyst.

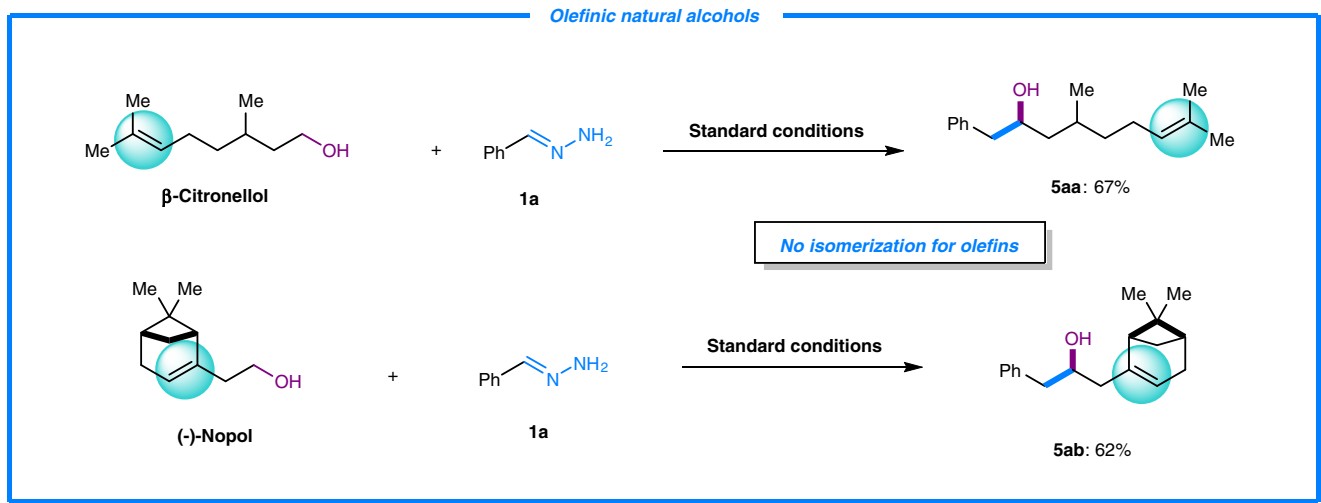

**Fig. 4 Reactivity of olefinic natural alcohols.** Standard reaction conditions: **1a** (0.6 mmol), **2a** (0.2 mmol), Ru(PPh$_3$)$_3$Cl$_2$ (5 mol %), ligand (5 mol %) and K$_3$PO$_4$ (2 equiv.) in 2-MeTHF at 70 °C under N$_2$ atmosphere for 24 h. See Supplementary Information (SI) for details. Some naturally available olefinic alcohols also reacted efficiently, in which the olefin did not isomerize with the alcohol being the only reaction site. These two examples demonstrated the synthetic potentials, attributed to the special tolerance to olefins by the ruthenium catalyst.

ruthenium center, which interacts with a hydride and hydrogen gas is released[41]. Concurrently, the 1,2- addition process via a Zimmerman–Traxler chair-like transition state **d** is completed as we proposed previously[27]. Finally, after the C–C bond formation and the release of N$_2$ gas, the desired product is formed with the regeneration of the catalyst for the next cycle.

## Discussion

In conclusion, an oxidant-free Ru(II)-PNP catalyzed Grignard-type reaction with alcohol as a carbonyl surrogate was successfully demonstrated. This reaction takes advantage of both the kinetically favored dehydrogenation process provided by a phosphine-amine ligand and the thermodynamic driving force of the 1,2-addition to carbonyls by hydrazone with Ru(II) catalysis. The development of this transformation marks an evolution in the Grignard-type reaction, wherein direct construction of C–C bonds are possible from various naturally abundant alcohols, with a tolerance for sensitive functional groups and further expanding Grignard-type reactions from an early-stage construction to late-stage modifications. Future work includes a more in-depth investigation of the application potentials as well as mechanistic studies for this alcohol-surrogated Grignard-type reaction.

## Methods

**General procedure for reaction in Table 1.** Ru(PPh$_3$)$_2$Cl$_2$ (0.01 mmol), dcypf (0.01 mmol), K$_3$PO$_4$ (0.22 mmol) and oxidant (0.4 mmol, 2 equiv)* were added into a V-shaped reaction tube equipped with a stir bar in the glovebox. Then, the reaction tube was sealed and moved out of the glovebox. After that, **1a** solution (prepared by the method described in SI, Procedure A, 0.22 mL, 0.25 mmol) was added first, followed by the addition of **2a** (25.0 μL, 0.2 mmol). The mixture was stirred for 24 h. Then, 1,3,5-trimethoxylbenzene (11.2 mg, 0.067 mmol) was added in the mixture as standard. Then, the solution was filtered by celite and concentrated to dryness. The crude mixture was diluted by CDCl$_3$ to run the $^1$H NMR test to determine the $^1$H NMR yield.

*For entry 2, the reaction tube was sealed before exposed to air for 5 min. For entry 3, after removing the reaction tube out of glovebox, it was charged with O$_2$ via 3 times vacuum-refill by oxygen balloon.

**General procedure for reaction in Table 2.** Ru(PPh$_3$)$_2$Cl$_2$ (0.01 mmol), ligand (0.01 mmol) and K$_3$PO$_4$ (0.4 mmol) were added into a V-shaped reaction tube equipped with a stir bar in the glovebox. Then, the reaction tube was sealed and moved out of the glovebox. After that, 0.5 mL 2-Me-THF was added and followed by the addition of corresponding amount of **1a** (prepared by the method described in SI, Procedure B) and **2a** (25.0 μL, 0.2 mmol). The mixture was stirred for 24 h under N$_2$ at 70 °C. After completion, the solution was filtered by celite and

concentrated to dryness. Then, 1,3,5-trimethoxylbenzene (11.2 mg, 0.067 mmol) was added in the mixture as standard. The crude mixture was diluted by CDCl$_3$ and the $^1$H NMR test was run to determine the $^1$H NMR yield.

**General procedure for reactions in Figs. 3 and 4.** Ru(PPh$_3$)$_3$Cl$_2$ (0.01 mmol), **L1** (0.01 mmol), K$_3$PO$_4$ (0.4 mmol) and solid substrates were added into a V-shaped reaction tube equipped with a stir bar in the glovebox. Then, the reaction tube was sealed and moved out of the glovebox. After that, 0.5 mL 2-Me-THF was added first, followed by the addition of liquid substrates. The mixture was stirred under 70 °C for 24 h. The reaction mixture was filtered through a celite plug and washed with 2–3 mL CH$_2$Cl$_2$. The solvent was removed by a rotary evaporator and the residue was purified by column chromatography on silica gel (using hexane and ethyl acetate as eluents) to give the pure product.

**General procedure for products 3as-3au, 3hv, and 3hw.** Ru-PNP-3 (0.01 mmol), K$_3$PO$_4$ (0.4 mmol) and solid substrates were added into a V-shaped reaction tube equipped with a stir bar in the glovebox. Then, the reaction tube was sealed and moved out of the glovebox. After that, 0.5 mL 2-Me-THF was added first, followed by the addition of liquid substrates. The mixture was stirred under 100 °C for 24 h. The reaction mixture was filtered through a celite plug and washed with 2–3 mL CH$_2$Cl$_2$. The solvent was removed by a rotary evaporator and the residue was purified by column chromatography on silica gel (using hexane and ethyl acetate as eluents) to give the pure product.

**General procedure for products 3ax-3az.** Ru-PNP-1 (0.01 mmol), K$_3$PO$_4$ (0.4 mmol) and solid alcohol (0.2 mmol, if applicable) were added into a V-shaped reaction tube equipped with a stir bar in the glovebox. Then, the reaction tube was sealed and moved out of the glovebox. After that, 0.5 mL 2-Me-THF was added first, followed by the addition of **1a** (0.6 mmol, 3 equiv) and liquid alcohols (0.2 mmol, if applicable). The mixture was stirred under 70 °C for 24 h. The reaction mixture was filtered through a celite plug and washed with 2–3 mL CH$_2$Cl$_2$. The solvent was removed by a rotary evaporator and the residue was purified by column chromatography on silica gel (using hexane and ethyl acetate as eluents) to give the pure product. Specifically, for **3ay**, the mixture was diluted with CDCl$_3$ and the $^1$H NMR test was run to determine a trace amount of the desired product.

**Procedure for Fig. 3 (3ks, 3ls).** Ru(PPh$_3$)$_3$Cl$_2$ (0.01 mmol), **L3** (0.006 mmol) and K$_3$PO$_4$ (0.4 mmol) were added into a V-shaped reaction tube equipped with a stir bar in the glovebox. Then, the reaction tube was sealed and moved out of the glovebox. After that, **1** solution (prepared by the method described in Procedure B, 0.55 mL, 0.6 mmol) was added. The mixture was stirred at 100 °C for 24 h. The reaction mixture was filtered through a celite plug and washed with 2–3 mL CH$_2$Cl$_2$. The solvent was removed by a rotary evaporator and mesitylene was added to the residue as internal standard. The mixture was diluted with CDCl$_3$ and the $^1$H NMR test was run to determine a trace amount of the desired product based on the standard spectrum from literature (see SI for details).

**Fig. 5 Proposed mechanism for the alcohol-surrogated Grignard-type reaction.** The proposed mechanism starts from the tetracoordinated ruthenium complex. Both the dehydrogenation step and a C–C bond formation step experience six-membered-ring transition states. Specifically, **a** active species to start the catalytic cycle; **b** six-membered-ring transition state for dehydrogenation; **c** ruthenium-hydride species after the dehydrogenation step; **d** transition state for 1,2-addition of hydrazone to carbonyl; **e** intermediate right after C–C bond formation; **f** intermediate after denitrogen which then followed by the formation of desired product.

## Data availability
The authors declare that the data supporting the findings of this study are available within the article and Supplementary Information file, or from the corresponding author upon reasonable request.

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

## Acknowledgements

We acknowledge the Canada Research Chair Foundation (to C.-J.L.), the CFI, FQRNT Center for Green Chemistry and Catalysis, NSERC, the Killam Research Fellow by the Canadian Council of Arts and McGill University for support of our research. We also thank Y. Kim and S. Luo for proof reading, D. Farajat for language polishing.

## Author contributions

C.-C.L. discovered the reaction. C.-C.L., Z.Q., and Z.-P.C. developed the Ru-PNP catalytic system. C.-C.L. designed and conducted the experiments with the assistance of Z.Q. and Z.-P.C. H.W., M.M.S., and R.Z.K. did or helped doing the DFT calculation. C.-C.L. completed the manuscript with the assistance of Z.Q. and Z.-P.C. C.-J.L. guided the whole project and reviewed the manuscript.

## Competing interests

The authors declare no competing interests.
