## [Peer Review File · Nature Communications]

REVIEWER COMMENTS

Reviewer #1 (Remarks to the Author):

The manuscript by Li et al presents a novel catalytic Grignard-type C-C bond formation between a hydrazone, acting as a traceless carbanion synthon, and an in situ generated carbonyl from alcohols. The authors demonstrate a reasonable scope as well as suggesting a mechanism. As such, I find the work impressive and of more general interest. Generally, the manuscript is written in a reasonably scholar manner.

However, previous publications by the authors themselves (eg references 27 and 29 in the manuscript) as well as from others (eg reference 37 in the manuscript and Eur. JOC 2007, 5629) already showcase the hydrazone acting as a traceless carbanion synthon for the catalysed reaction with carbonyls or alcoholic carbonyl precursors. Furthermore, acceptorless alcohol dehydrogenation is a highly developed field. Hence, I cannot recommend publication in such a highly renowned journal as Nature Communications.

Furthermore, I have several suggestions to the authors:

- 1) In my mind, the scope does not really challenge a traditional Grignard reaction, with the exception of the N-Boc protected substrate. I suggest to test eg amides, esters, carbonates, nitrils, etc.
- 2) The manuscript would benefit greatly from isolating and characterizing the (assumed) catalytically active complex.
- 3) The optimised conditions seem to be adopted from previously published projects. This work might benefit from a (minor) new optimisation protocol. For example, acceptorless alcohol dehydrogenation is typically greatly improved by intensely refluxing the media. Also, testing other metal precursors is relevant and might aid the mechanistic considerations.
- 4) The mechanism should be founded as well as argued based on the experimental results in the work. As of now, the proposed mechanism is claimed without any references to their own observations.
- 5) In my mind, some citations are missing: When mentioning PNP-ligands: One of more of Beller, *ACIE* 2011, 50, 9593; Gusev, *Org. Met.* 2011, 30, 3479; Schneider, *Inorg. Chem.* 2010, 49, 5482. When discussing the reactivity of the cyclobutyl substrate, perhaps add a reference substantiating your arguments.
- 6) There are several typos. For example, the abbreviations are inconsistent and the references need a revision.

Reviewer #2 (Remarks to the Author):

Li and co-workers have reported a sustainable alternative to the Grignard reaction for the construction of secondary and tertiary alcohols. This work is based on their recent discovery that hydrazone can be employed as traceless carbanion equivalent and their couplings with ketones and aldehydes led to the formal Grignard reaction (*Nat. Chem.* 2017, 9, 374).

Here, instead to use ketones or aldehydes, they start from alcohols using an in-situ dehydrogenation process. They are several advantages to use alcohols as starting materials: higher stability, natural feedstocks, and availability. Unlike to previous tandem oxidative processes, this reaction required only one catalyst. The PNP-Ru can achieve both alcohol dehydrogenation and the umpolung addition process. The scope of the reaction has been studied in detail and is very general, including the use of natural olefinic alcohols without isomerization or reductions.

Overall, this manuscript deserves to be published after answer these minor concerns:

1. Table 2, in the general equation: Dcypf should be replaced by L
2. Why did the authors not employ a well-defined catalyst (type complex A) rather than in-situ preparation? Is there a reactivity difference?

Reviewer #3 (Remarks to the Author):

In this article, Chao-jun Li et al reported a Ru-complex catalyzed dehydrogenative coupling reaction with alcohols as carbonyl surrogates and aldehyde hydrazones as carbanion equivalent. This strategy allows to access Grignard-type products with naturally abundant alcohols and aldehydes as starting materials. In my opinion, this new unique Grignard-type reaction with alcohols as carbonyl surrogates is highly interested to organic researchers. It may deserve publication in Nature Communication, but several points require attention:

1. I have some concerns regarding the scope of the process. From a synthetic point of view, this reaction shows some limitations regarding the scope. Only simple alcohols and benzaldehyde hydrazones are compatible with the process. Secondary alcohols and aliphatic aldehyde hydrazones proved to be less active. Although the authors have made an effort to demonstrate the synthetic application of this transformation by applying two naturally occurring complex alcohols (Scheme 4), more examples regarding more kind of aldehyde hydrazones and alcohols should be concluded.
2. In Table 2, only nine different conditions with three ligands were tested. It's necessary to perform a more detail study on condition-optimization. More ligands and catalytic systems should be tested (For examples, the ligands and catalytic systems reported in *Organometallics* 2016, 35, 2840–2849; *Nat. Commun.*, 2015, 6, 6859; *Chem. Eur. J.*, 2008, 14, 10201; et al) to get a more general condition which may be helpful to broaden the reaction scope.
3. In this reaction, 3 equiv. amount of aldehyde hydrazones 1a and its analogues is required due to the instability of this compound. Stable aldehyde hydrazones, for example these derived from aldehydes and aryl hydrazines, maybe helpful to reduce the dosage of the hydrazone substrates. Also, a stable aldehyde hydrazones can tolerate a higher reaction temperature which apparently can accelerate the dehydrogenation process according literature reports, thus leading to a better reaction scope. Therefore, some stable aldehyde hydrazones should be tested.
4. In Table 2 and Scheme 3, the reaction time and the amount of K₃PO₄ should be noted. In the chemical scheme of Table 2, the term "dcypf" (upon the second arrow) should be deleted. Some of the yields in SI are inconsistent with the yields given in manuscript (3af, 3ag, 3ap). The authors should carefully re-check their manuscript to avoid such kind of mistakes.

Our point-to-point responses

Reviewer #1 (Remarks to the Author):

Comments: The manuscript by Li et al presents a novel catalytic Grignard-type C-C bond formation between a hydrazone, acting as a traceless carbanion synthon, and an in situ generated carbonyl from alcohols. The authors demonstrate a reasonable scope as well as suggesting a mechanism. As such, I find the work impressive and of more general interest. Generally, the manuscript is written in a reasonably scholar manner.

However, previous publications by the authors themselves (eg references 27 and 29 in the manuscript) as well as from others (eg reference 37 in the manuscript and Eur. JOC 2007, 5629) already showcase the hydrazone acting as a traceless carbanion synthon for the catalysed reaction with carbonyls or alcoholic carbonyl precursors. Furthermore, acceptorless alcohol dehydrogenation is a highly developed field. Hence, I cannot recommend publication in such a highly renowned journal as Nature Communications.

Furthermore, I have several suggestions to the authors:

Our response: We are grateful for reviewer's comments. It is true that acceptorless dehydrogenation of alcohol is a well studied field. However, in our reaction, we believe that the most meaningful part is that it could realize the synergistic relay with the 1,2-addition of hydrazone with the assist of ruthenium catalyst, which is extremely difficult to realize by other reported methods. Moreover, the acceptorless dehydrogenation of alcohol usually requires high temperature (higher than 100 °C) and usually less efficient to primary alcohols. In our case, the utilization of this synergistic relay somewhat solved the historical challenge. Consequently, we believe that this work can raise interest by the broad audience in chemistry. Also, in the real synthetic case, carbonyl is less compatible and difficult to access than alcohols. Thus, empowering alcohol as aldehyde precursor has a good synthetic application potential. In our revised manuscript, we have adopted the suggestions that the reviewer pointed out. Thus, we are hope the reviewer could kindly reconsider our work.

Question: In my mind, the scope does not really challenge a traditional Grignard

reaction, with the exception of the N-Boc protected substrate. I suggest to test eg amides, esters, carbonates, nitrils, etc.

Our response: We are grateful for the reviewer's kind suggestions. We have tried some substrates with some sensitive functional groups and find that most of them can be successfully tolerated under the reaction system such as amides and esters, However, to be noticed, carbonates has a poor tolerance maybe due to its high leaving ability to cause some side reactions such as oxidative addition with ruthenium. While nitriles are not a good substrate in our system maybe due to its high coordinating ability. We will continue this work with other catalytic systems in future studies.

Question: The manuscript would benefit greatly from isolating and characterizing the (assumed) catalytically active complex.

Our response: That is a very good suggestion. We have tried our best to pre-synthesize the well-defined Ru-PNP complex and used them to conduct our reaction. It does not actually show a big difference with adding catalyst and ligand in-situ. However, we do benefit from using the well-defined catalyst in conducting the reaction with functional group containing substrates. This catalyst helped us to obtain a moderate to high yield with amide or ester-containing substrates.

Question: The optimised conditions seem to be adopted from previously published projects. This work might benefit from a (minor) new optimisation protocol. For example, acceptorless alcohol dehydrogenation is typically greatly improved by intensely refluxing the media. Also, testing other metal precursors is relevant and might aid the mechanistic considerations.

Our response: We are grateful for the reviewer's kind suggestions. Actually, we have tried many different metal precursors early in our work of 1,2-addition to carbonyl with hydrazones. However, only ruthenium catalyst shows good reactivity. In our reaction, the catalyst needs to be efficient both with dehydrogenation and 1,2-addition and in this case, ruthenium is proved to be the best choice. We have tried rhodium or iridium catalyst for this reaction but none of them give us optimized results. Also, we have tried higher temperature for this reaction but got even lower yield, possibly

because the high temperature favors Wolff-Kishner reduction and consumes the hydrazone substrates.

Question: The mechanism should be founded as well as argued based on the experimental results in the work. As of now, the proposed mechanism is claimed without any references to their own observations.

Our response: We are grateful for the reviewer's kind suggestions. Actually, this reaction includes a dehydrogenation process and a 1,2-addition process. For the latter one, we have done some studies in our previous work and we have already cited in our original manuscript. Also, we proposed the Noyori-typed six-membered ring transition state for the dehydrogenation, based on the fact that only PNP-pincer ligand works efficiently in our reaction system, which suggests that nitrogen is vital to this reaction (mostly dehydrogenation process). To better illustrate this proposal, we did a DFT calculation for the dehydrogenation step. The resulted six-membered-ring transition state is a reasonable transition state which is support by IRC study. The mechanism also expalins why the PNP-pincer type ligand can synergistically drive the dehydrogenation and 1,2-addition process. Furthermore, in our revised manuscript, for every single step of the mechanism, we cited the related reference (if applicable).

Question: In my mind, some citations are missing: When mentioning PNP-ligands: One of more of Beller, ACIE 2011, 50, 9593; Gusev, Org. Met. 2011, 30, 3479; Schneider, Inorg. Chem. 2010, 49, 5482. When discussing the reactivity of the cyclobutyl substrate, perhaps add a reference substantiating your arguments.

Our response: We are grateful for reviewer's kind suggestions. We have added those references in our revised manuscript.

Question: There are several typos. For example, the abbreviations are inconsistent and the references need a revision.

Our response: We are sorry for our mistake. We have corrected those inconsistent abbreviations.

Reviewer #2 (Remarks to the Author):

Comments: Li and co-workers have reported a sustainable alternative to the Grignard reaction for the construction of secondary and tertiary alcohols. This work is based on their recent discovery that hydrazone can be employed as traceless carbanion equivalent and their couplings with ketones and aldehydes led to the formal Grignard reaction (Nat. Chem. 2017, 9, 374).

Here, instead to use ketones or aldehydes, they start from alcohols using an in-situ dehydrogenation process. There are several advantages to use alcohols as starting materials: higher stability, natural feedstocks, and availability. Unlike to previous tandem oxidative processes, this reaction required only one catalyst. The PNP-Ru can achieve both alcohol dehydrogenation and the umpolung addition process. The scope of the reaction has been studied in detail and is very general, including the use of natural olefinic alcohols without isomerization or reductions.

Overall, this manuscript deserves to be published after answer these minor concerns:

Our response: We are grateful for the reviewer's appreciation. We have revised our manuscript accordingly

Question: Table 2, in the general equation: Dcypf should be replaced by L

Our response: We are sorry for the mistake and have corrected them in the revised manuscript.

Question: Why did the authors not employ a well-defined catalyst (type complex A) rather than in-situ preparation? Is there a reactivity difference?

Our response: We are grateful for the reviewer's kind suggestions. We have pre-synthesized the well-defined catalyst Ru-PNP and tested them in our reaction. The result did not show significant difference with in-situ prepared one. Due to

inconvenience of synthesizing the pure catalyst, we kept the in-situ preparation as our optimized condition for most substrates. However, with some functional group-containing substrates, the pre-synthesized catalyst was indeed beneficial, and we used the well-defined catalyst for those substrates in the revised manuscript.

Reviewer #3 (Remarks to the Author):

Comments: In this article, Chao-jun Li et al reported a Ru-complex catalyzed dehydrogenative coupling reaction with alcohols as carbonyl surrogates and aldehyde hydrazones as carbanion equivalent. This strategy allows to access Grignard-type products with naturally abundant alcohols and aldehydes as starting materials. In my opinion, this new unique Grignard-type reaction with alcohols as carbonyl surrogates is highly interested to organic researchers. It may deserve publication in Nature Communication, but several points require attention:

Our response: We are grateful for the reviewer's appreciation. We have revised our manuscript accordingly

Question: I have some concerns regarding the scope of the process. From a synthetic point of view, this reaction shows some limitations regarding the scope. Only simple alcohols and benzaldehyde hydrazones are compatible with the process. Secondary alcohols and aliphatic aldehyde hydrazones proved to be less active. Although the authors have made an effort to demonstrate the synthetic application of this transformation by applying two naturally occurring complex alcohols (Scheme4), more examples regarding more kind of aldehyde hydrazones and alcohols should be concluded.

Our response: We are grateful for the reviewer's kind suggestions. Under our original conditions, the substrate scope is indeed limited to primary alcohols. In our revised manuscript, we find a less bulky PNP ligand which could greatly improve the reactivity of secondary alcohols. Thus, we added examples of secondary alcohols in our revised manuscript, which can reach moderate to high yields. Furthermore, for primary alcohols, we added some substrates with sensitive functional groups such as

esters and amides. However, with the hydrazone part, we have tried our best to find an efficient system for the aliphatic aldehyde hydrazones. However, aliphatic aldehyde hydrazones showed much lower reactivity under PNP catalytic system. The main reason for that, we proposed, is the instability of the anion intermediate (without the conjugation of adjacent aryl group). That could be the inherent limitation of this type of reaction at this time. However, we are continuously carrying out more research to empower aliphatic aldehyde hydrazones to undergo this dehydrogenative Grignard reaction.

Question: In Table 2, only nine different conditions with three ligands were tested. It's necessary to perform a more detail study on condition-optimization. More ligands and catalytic systems should be tested (For examples, the ligands and catalytic systems reported in *Organometallics* 2016, 35, 2840–2849; *Nat. Commun.*, 2015, 6, 6859; *Chem. Eur. J.*, 2008, 14, 10201; et al) to get a more general condition which may be helpful to broaden the reaction scope.

Our response: We are grateful for reviewer's kind suggestions. In our revised manuscript, we have added more kinds of pincer-type ligands, which we have in our lab or easily accessible from vendor such as PNN type and $\text{PN}_{\text{pyridine}}\text{P}$ type ligand which is potentially reactive for dehydrogenation. However, none of those ligands showed greater reactivity than **L3** as what we used. However, to be noticed, a less bulky PNP ligand such as **L-Et** in our revised manuscript did show a greater reactivity for secondary alcohols. This trend suggested that the methyl protected PNP ligand, which is even less bulky, might gave a greater efficiency. Unfortunately, because of Covid-19 pandemic, the starting material for the synthesis of methyl-substituted-PNP ligand is not available anywhere in the world as we tried to order. We will continue to investigate the methyl one in our future study once the pandemic is over. On the other hand, the less bulky **L-Et** ligand did not show great difference with **L3** when catalysing the reaction of primary alcohols. Thus, for primary alcohols, we still use **L3** as the optimized condition due to its much easier accessibility and lower costs.

Question: In this reaction, 3 equiv. amount of aldehyde hydrazones 1a and its analogues is required due to the instability of this compound. Stable aldehyde

hydrazones, for example these derived from aldehydes and aryl hydrazines, maybe helpful to reduce the dosage of the hydrazone substrates. Also, a stable aldehyde hydrazones can tolerate a higher reaction temperature which apparently can accelerate the dehydrogenation process according literature reports, thus leading to a better reaction scope. Therefore, some stable aldehyde hydrazones should be tested.

Our response: We are grateful for the reviewer's kind suggestions. We have tried some stable aldehyde hydrazones (substituted hydrazones) but they are not active substrates in our reaction. We proposed that when hydrazone serves as carbanion equivalent, based on its inherent reactivity, two of the protons on the nitrogen are all necessary. That is why only non-substituted hydrazones work.

Question: In Table 2 and Scheme 3, the reaction time and the amount of K₃PO₄ should be noted. In the chemical scheme of Table 2, the term "dcypf" (upon the second arrow) should be deleted. Some of the yields in SI are inconsistent with the yields given in manuscript (3af, 3ag, 3ap). The authors should carefully re-check their manuscript to avoid such kind of mistakes.

Our response: We are grateful for the reviewer's kind suggestions. We are sorry for our mistake. In our revised manuscript, we have corrected all the mistake mentioned above.

REVIEWERS' COMMENTS

Reviewer #1 (Remarks to the Author):

The authors have addressed all my concerns with a comprehensive piece of work. In my mind, the manuscript now contains several more interesting aspects.

Reviewer #2 (Remarks to the Author):

In the revised manuscript by Chao-Jun Li most of the reviewer concerns have been addressed.

1. The substrate scope has been extended to a couple of substrates bearing a more sensitive functional group (esters alone, carbamate).
2. Well-defined pincer catalyst has been tested and shown for some substrates better activities.
3. The reaction has been extended to secondary alcohols using less bulky PNP ligand.

This new data enhance the impact of this manuscript and can now be accepted as it is.

Reviewer #3 (Remarks to the Author):

The authors well answered my concerns, and provided their new data in the revised manuscript. I do agree the revised manuscript to be published on Nature Communications.